# A New Path of Quench-Induced Residual Stress Control in Thick 7050 Aluminum Alloy Plates

**Shengping Ye [1,3], Kanghua Chen [1,2,3,*], Changjun Zhu [1,2,3,*] and Songyi Chen [1,2,3]**

[1]   Collaborative Innovation Center of Advance Nonferrous, Central South University, Changsha 410083, China; yeshengping@csu.edu.cn (S.Y.); sychen08@csu.edu.cn (S.C.)

[2]   State Key Laboratory of High Performance Complex Manufacturing, Central South University, Changsha 410083, China

[3]   State Key Laboratory of Powder Metallurgy, Central South University, Changsha 410083, China

*   Correspondence: khchen@csu.edu.cn (K.C.); zhucj96@csu.edu.cn (C.Z.); Tel.: +86-0731-8883-0714 (K.C.); Fax: +86-0731-8871-0855 (K.C. & C.Z.)

**Abstract:** The high magnitude of quench-induced residual stress in thick aluminum plates is attributed not only to high thermal stress but also to high yield strength due to quench-induced precipitation hardening. To date, lowering the thermal stress is the only path to reduce the residual stress in the design of quenching technology. In this paper, a new path is proposed that reduces the residual stress through decreasing the yield strength at ambient temperatures by eliminating the precipitation hardening effect during quenching. As certified in several experiments, the high yield strength of thick as-quenched plates decreases rapidly from a short period of extra heat preservation at relatively higher temperatures. Therefore, an interrupted quenching method is proposed, wherein quenching is interrupted after an initial cooling period and the sample is placed in air to make the temperature field uniform; afterward, the sample is cooled to room temperature. Interrupted quenching tests were conducted on 115 mm thick 7050 aluminum plates and significant residual stress reductions were observed in the specimens compared with the residual stresses in the specimens subjected to regular quenching.

**Keywords:** 7050 alloys; quenching; residual stress; precipitation; clustering

## 1. Background

Aluminum alloy 7050 is a heat-treatable alloy that achieves high mechanical strength through age-hardening heat treatments and is widely used in the aircraft and aerospace industries [1,2]. Prior to age hardening, aluminum alloys are prepared by solution heat treatment and quenching. The objective of the solution heat treatment is to dissolve the secondary phases into the aluminum matrix at a particular temperature, and the objective of quenching is to cool the material to room temperature as quickly as possible to obtain a supersaturated solution. Then, the supersaturated solution will form dispersed strengthening particles in the matrix during aging, which improves mechanical properties [3]. The quenching step is essential for the final mechanical properties and is required to balance two contradictory effects, namely, the quench-induced coarse precipitation with sizes of approximately 100 nm and the residual stress resulting from thermal gradients formed during quenching. Residual stress is known to have detrimental effects on the final mechanical properties and deteriorate manufacturing accuracy, especially for thick plates [4]. The elimination and inhibition of residual stress is necessary, thus, quenching is followed by stress relief. However, residual stress cannot be removed completely in the stress relief step.

In addition to the stress relief step after quenching, the regulation of the quenching step itself has received attention. It is well known that decreasing the cooling rate during quenching can lower the

magnitude of residual stress; however, a lower cooling rate may produce additional quench-induced coarse precipitations, which affects the final mechanical properties. In most cases, ensuring the finial mechanical properties has a higher priority than residual stress reduction, and thus residual stress control is difficult due to the limited regulation window of cooling rate during quenching. According to the principle of precipitation during quenching for most 7xxx aluminum alloys, a high cooling rate must be maintained within the quench-sensitive temperature range, which typically ranges from 200 °C to 400 °C. If this condition is not met, significant additional performance losses will occur [5]. In a recent study, a multistage quenching process was proposed for 120 mm thick AA 7050 plates based on spray quenching technology that adopted a high cooling rate during the initial stage until cooling to temperatures less than 200 °C, and a lower cooling rate was adopted in the second stage to control the residual stress [6]. Compared with regular quenching, multistage quenching can reduce the maximum residual stress by only approximately 5–10%. On the premise of ensuring hardenability, it seems that a higher residual stress reduction is difficult to acquire because the cooling rate in temperatures below the quench-sensitive temperature range do not affect the residual stress as much as the cooling rate in the quench-sensitive temperature range.

Furthermore, the material will harden by the precipitation phenomena that occur during quenching. Hardening during the quench will give rise to a higher yield strength at ambient temperatures during quenching, which in turn will produce larger residual stress magnitudes [7]. This hardening effect is especially significant in the quenching of thick plates. For example, the yield strength of as-quenched thick AA7449 plates, wherein the 75 mm thick plate exhibited a much larger yield strength (255 MPa) than the 20 mm thick plate (210 MPa) [8]. These additional hardening effects make it more difficult to reduce the residual stress by lowering the cooling rate.

For thick plates, it is conceivable that the hardening effects that accumulate in a relatively long cooling process must be more significant than those observed in thin plates, and these hardening effects are partly responsible for the high-magnitude residual stresses in thick plates. These hardening effects were first ascribed to the homogeneous nucleation of the $\eta'$ phase, such as the low-temperature precipitation that occurs during fast cooling from 250 to 150 °C in a commercial AA7150 alloy [9]. However, it is doubtful that the uniform nucleation in such a short quenching time can support the remarkable hardening effects. In a more recent study, the in situ small-angle X-ray scattering (SAXS) measurements demonstrated that the nanosized precipitates formed during quenching in an AA7449 alloy harden the material significantly, and these authors suggested that the hardeners formed during quenching are mainly solute clusters, and that other larger precipitates can be neglected [10]. It should be noted that a special phenomenon was exhibited during the experiment, wherein the volume fraction of accumulated clusters after quenching decreased significantly after reheating to temperatures between 80 to 150 °C [11]. This phenomenon can be explained as follows: The solute clusters formed in such short times during quenching that they were supercritical nucleated and were not as stable as the larger precipitates were. Referred to the kinetic Monte Carlo simulation of clustering in AA7050, the early-stage small clusters in the simulation are loose aggregates that were regarded as unstable because they appeared and disappeared easily [12]. Therefore, the quench-induced clusters are easily dissolved upon heating, which will lead to a reduction in the yield strength of a hardened material after heating to relatively higher temperatures. For example, in an AA6082 alloy, the dissolution of clusters may lead to a marked decrease in the Vickers hardness within the first minute of artificial aging at temperatures greater than 210 °C [13].

It can be inferred that the quench-induced residual stress can be reduced not only by reducing the thermal stress but also by eliminating the high level of yield strength of a significantly hardened material (corresponding to the as-quenched aluminum plates). As previously mentioned, the high yield strength of an as-quenched material can be rapidly decreased by reheating to a relatively higher temperature.

Therefore, an optimized quenching technology called interrupted quenching was proposed, which follows a new path to reduce the residual stress by lowering the yield strength of the material rather

than just lowering the thermal stress. In this new path, after the core of the part was cooled down to quench-sensitive temperatures (210 °C), the cooling is stopped and the part is placed in air for a period of time so that the surface temperature reheated to 150–160 °C to make the accumulated hardeners (solute clusters) partly dissolve during quenching. This process produces a relative low yield strength, which reduces the residual stress. Afterward, the part is cooled to room temperature.

Interrupted quenching tests were conducted for 115 mm thick 7050 aluminum plates by using spraying quenching equipment, and the samples subjected to interrupted quenching were compared with samples subjected to regular (spray) quenching. The results show that this interrupted quenching approach has a satisfactory effect on residual stress reduction.

## 2. Experiments

### 2.1. Materials

The objective alloy was a commercial high-strength aluminum alloy, namely, AA7050, which has the composition shown in Table 1. The quenching experiment was performed on rectangular samples with dimensions of 250 (R) mm × 250 (W) mm × 115 (T) mm that were obtained from a 115 mm thick hot rolled aluminum plate with 8000 (R) mm × 800 (W) mm × 115 (T) mm, wherein they were sampled from the middle part of the width direction, as shown in Figure 1a. In this Figure, R is the rolling direction, W is the width direction, and T is the thickness direction. The rectangular sample was solution heat treated at 476 °C for 1 h and cooled (quenching) with spray quenching equipment. The crack compliance method (CCM) was applied after quenching to measure the as-quenched residual stress, and the sample was cut in half perpendicular to the width direction for the CCM measurement. Then, cylindrical samples were machined 15 mm in length and 10 mm in diameter, sampling position is shown in Figure 1b,c. The longitudinal axis of each sample was parallel to the width direction. The hardness and electrical conductivity values were measured using the cylindrical samples after aging treatment at T74 (120 °C, 6 h + 170 °C, 12 h).

**Table 1.** Chemical composition of 7050 aluminum alloy (wt.%).

| Zn | Mg | Cu | Zr | Fe | Si | Cr | Mn | Ti | Al |
|---|---|---|---|---|---|---|---|---|---|
| 6.27 | 2.37 | 2.21 | 0.11 | 0.11 | 0.11 | 0.03 | 0.05 | 0.05 | Bal. |

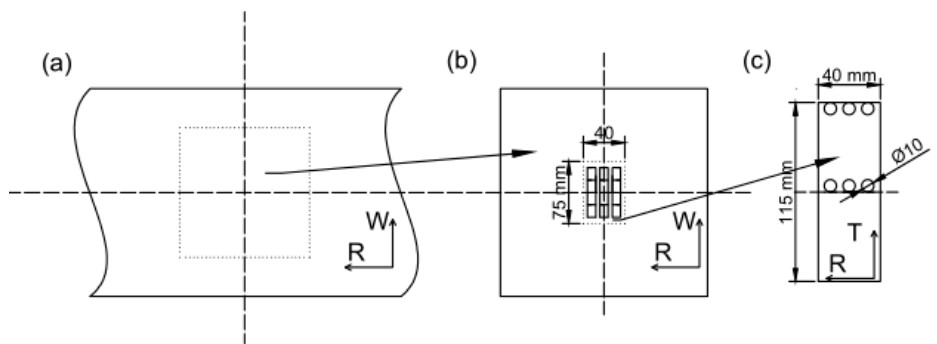

**Figure 1.** Schematic diagrams of the sampling positions.

### 2.2. Spray Quenching and Equipment

Each sample with dimensions of 250 (R) × 250 (W) × 115 (T) mm was solution heat treated at 476 ± 2 °C for 1 h in a resistance-heated furnace. After solution heat treatment, the plate was moved to the spraying equipment by a track. The spray quenching equipment contains a piping, spray nozzle, water purifier and pump. The sample was placed vertically in both the furnace and spray quenching equipment, as shown in Figure 2. The water pressure was set to 500 kPa, and the flux density was set to 108 L·m$^{-2}$·s$^{-1}$, which was regulated by the power of the pump, and measured by the hydraulic

indicator and flowmeter equipped in the pipe. The spraying was started or stopped within 2 s by turning on or turning off the pump.

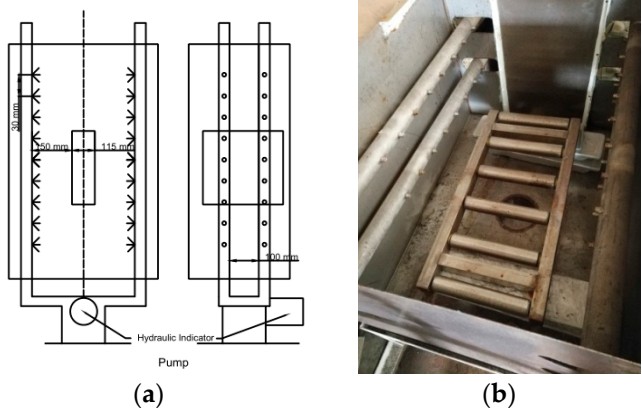

(**a**)          (**b**)

**Figure 2.** (**a**) Schematic and (**b**) photo of the spray quenching equipment.

Two quenching experiments were conducted, including interrupted quenching and comparison experiments with regular quenching. The interrupted quenching is: Interrupting the spray quenching after 30 s, placing the sample in air for 20 s and then spray quenching to room temperature. The comparison experiment involved directly spray quenching to room temperature. When the spraying was stopped, it is likely that the surface temperature will increase again due to the higher temperature in the core of the sample.

*2.3. Residual Stress Measurement*

The crack compliance method (CCM) was used to measure the residual stress in both samples. The measuring principle of the CCM is as follows: To release the residual stress, a crack of increasing depth was induced on the surface of the measured object, and the residual stress was calculated by measuring the strain at a specific position, as shown in [14]. In this study, the CCM measurement is performed on a wire cutting machine, wherein the wire diameter is 0.18 mm, the feed rate is 1 mm/min, and the crack width is approximately 0.4 mm. The strain data were recorded by a CML-1H static strain instrument (Qinhuangdao, China) and recorded per 5 mm depth. The microstrain data were measured by BE120-5AA resistance strain gauges (Nanjing, China) with a sensitive gate size of 5 mm × 3 mm. The positions of the strain gauges are shown in Figure 3. Five gauges were used for residual stress measurement, and the residual stress was measured along thickness direction (cut in the middle of rolling direction as shown in Figure 3b).

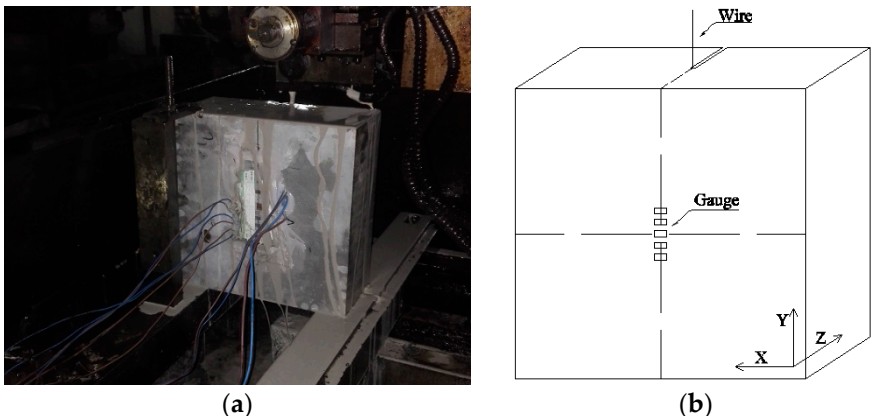

(**a**)          (**b**)

**Figure 3.** (**a**) Photo and (**b**) schematic view of the Crack Compliance Method (CCM) measurement setup.



*2.4. Compression Tests*

Precipitation (clustering) may have stronger effects on some mechanical properties compared with parameters weakly affected by precipitation (elastic modulus, Poisson's ratio and coefficient of thermal expansion); other parameters, especially the yield strength, are most strongly affected by precipitation. In this study, the yield strength was derived from the true flow stress, which is easily plotted from the data recorded by the thermal simulator during hot compression tests. Compression tests were conducted at room temperature (25 °C) with a strain rate of 0.001 s$^{-1}$. The yield strength was recorded as the value of $\sigma_{0.02}$ (corresponding to the true stress in 0.2% plastic strain) inferred from the true strain–stress curves in compression tests.

The compression specimens have dimensions of $\Phi$ 10 $\times$ 15 mm, and the length direction of each compression specimen was parallel to the width direction of the thick plate. The samples were taken from the surface and core area in the thickness direction, as shown in Figure 1c. Three tests were conducted for each value and the average was taken as the yield strength.

*2.5. Hardness Testing*

The hardness was measured along the thickness direction per 5 mm in the center of the rectangle sample, performed by a Vickers Hardness Tester HV-50 produced by Shanghai union Test Equipment Co., Ltd., (Shanghai, China) and calibrated according to GB/T 231.2-2012. Hardness samples was stored at a temperature below 25 °C.

*2.6. Electrical Conductivity Measurements*

Electrical conductivity testing was performed after quenching and after T74 aging treatment. The upper and lower surfaces of each cylindrical sample were measured three times and the average value was taken (both upper and lower surfaces). Electrical conductivity measurements were conducted by a portable electrical conductivity device produced by Shanghai Kuosi Electronic Co., Ltd. (Shanghai, China), which was calibrated in accordance with GB 11007-1989.

*2.7. Transmission Electron Microscopy*

Slices cut from cylindrical samples from surface and core (cut section was perpendicular to the longitudinal axis of each sample), respectively. Samples were mechanically ground to a thickness of approximately 150 nm from which discs of 3 mm diameter were punched. The material was in the artificial aged (T74) condition. These discs were then electro-polished to perforation using a Struers Tenupol-5 in a solution of 22% HNO3 and 78% CH3OH at $-30$ °C. Bright-field images were taken using a JEOL 2011 transmission electron microscope (JEOL Ltd., Tokyo, Japan) operating at 200 kV.

## 3. Results and Discussion

In this study, samples from the surface and core in different states were studied, as shown in Figure 4a. The heat treatment status for each sample is shown in Table 2. The results show that the as-quenched stage both in the surface and core exhibit a significantly higher yield strength (226/216 MPa) compared with the samples subjected to an additional solution heat treatment at 476 °C for 30 min (175/176 MPa). This finding indicates that the as-quenched material was significantly hardened by precipitation or clustering during quenching. Furthermore, the samples that were subjected to extra reheating at 160 °C for 20 s (in a 160 °C salt bath) and then cooled to 25 °C exhibited a decrease in yield strength (192/194 MPa). In addition, the electrical conductivity is decreased upon heating too, as shown in Figure 4b. These findings indicate that the accumulated as-quenched hardeners (clusters) may dissolve upon heating because the structural stability of clusters are far less than those of large precipitates, as mentioned by the in situ SAXS results [11].

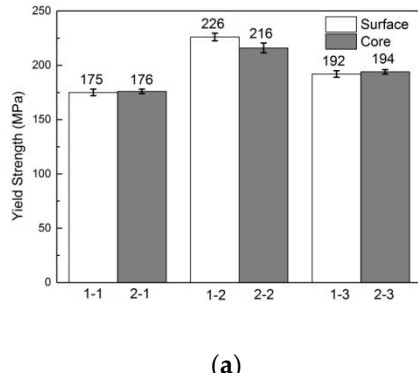
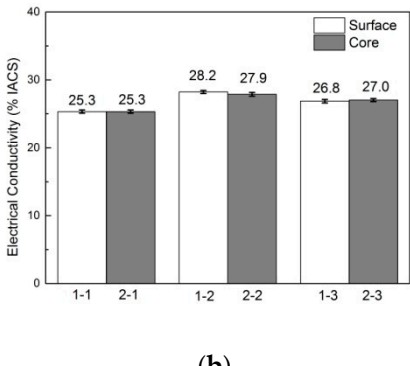

(**a**)　　　　　　　　　　　　　　　　　　　　　　(**b**)

**Figure 4.** (**a**) Yield strength and (**b**) Electrical conductivity of surface and center of 115 mm-7050 aluminum alloy plate with different heat treatments.

**Table 2.** Heat treatments for the compression samples.

| Number | Sampling Position | Heat Treatments |
|---|---|---|
| 1-1 | Surface | Additional solution heat treatment at 476 °C for 30 min |
| 1-2 | Surface | As-quenched |
| 1-3 | Surface | As-quenched, reheated to 160 °C for 20 s |
| 2-1 | Core | Additional solution heat treatment at 476 °C for 30 min |
| 2-2 | Core | As-quenched |
| 2-3 | Core | As-quenched, reheated to 160 °C for 20 s |

The mechanism of residual stress production has been well recognized in previous research, which is considered to be the result of inhomogeneous elastoplastic deformation due to the thermal stress during quenching [15–17]. The magnitude of the residual stress exhibits a linear trend with the magnitude of the residual plastic deformation. In most cases, the evolution of the elastoplastic deformation includes two main stages. The first deformation stage occurs during the initial portion of the quench, which is mainly driven by thermal stress and produces a tensile deformation in the surface and a compression deformation in the core. Taking the precipitation hardening effect into account, the increase in yield strength may hinder these deformations, but the accumulation of the hardening effect over such a short time is usually not sufficient to significantly affect the deformation. The second deformation stage occurs in the latter portion of the quench, when the thermal stress decreases to a certain level and the internal elastic stress caused by the former plastic deformation exceeds the yield strength at ambient temperatures and produces a deformation contrary to the deformation that occurs in the first stage. The plastic deformation in the second stage should be promoted because this deformation may decrease the final magnitude of the residual stress. The hardening effect in the second stage increases the yield strength of the material, which increases the difficulty of plastic deformation in the second stage. It is possible that decreasing the hardening effect may reduce the residual stress.

Therefore, an interrupted quenching technology was proposed based on the idea of reducing the yield strength through an extra heating process aiming to promote plastic deformation in the second stage. That is, after quenching for a period of time (30 s), the surface was cooled to approximately 100 °C and the core was cooled to about 200 °C (below the quench-sensitive temperature range). Then, the spray quenching was interrupted for a period of time (about 20 s). In interrupting, the heat transfer from surface to the surroundings decreases significantly, and the surface material will be re-heated by the heat transfer from the internal to surface. Finally, the material was cooled to room temperature through spraying.

The sketch of the interrupted quenching is shown in Figure 5, and includes the time–temperature–property (TTP) diagram of 7050 aluminum alloy [5] and the cooling curves of surface and core in thickness direction in the middle-part of samples undergoing regular quenching and interrupted quenching, respectively. The TTP diagram was corresponding to the 99.5% maximum

hardness that approximately reflects the critical time of precipitation beginning. The cooling curves corresponding to spray quenching tests were provided by the finite element analysis (FEA) developed using ABAQUS [18]. The parameters used in the heat transfer analysis were consistent with the parameters in literature [6], the starting geometry was chosen as an isotropic rectilinear block measuring 250 mm $\times$ 250 mm $\times$ 115 mm, and the initial temperature was set to 476 °C. The TTP diagram suggests that the regulation window is limited and that the cooling rate can only be adjusted when the core temperature decreases below 200 °C. According to the core cooling curve of the 115 mm thick rectilinear block, a high cooling rate must be sustained in initial 30 s (approximation) until temperature decreases below the quench-sensitive temperature range. Further, the tendency of a temperature re-rise is shown in the surface cooling curve from the FEA calculations, and the measurement of surface temperature using a contact thermocouple in the interrupting period shows that the temperature may re-rise to 141 °C within 20 s (as shown in the square dots in Figure 5).

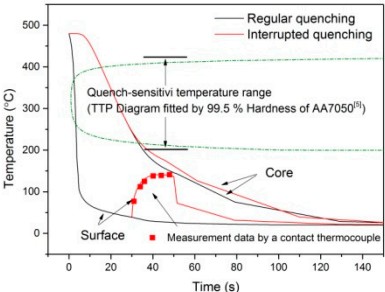

**Figure 5.** The sketch of interrupted quenching technology.

A high cooling rate was adopted during the initial quenching to ensure the hardenability of the material, and an adjustment was made to the cooling rate only in the latter stage of quenching to avoid additional coarse precipitation. For the surface, a large number of hardeners accumulated, which created a significant hardening effect. As previously mentioned, the hardeners may dissolve easily upon heating resulting in a lower yield strength. This phenomenon leads to a decrease in the yield strength at ambient temperatures, thus promotes the second-stage plastic deformation. In addition, the thermal stress, as the resistance of the second stage of plastic deformation, may also decrease by the tendency to a uniform temperature field. In the interrupting period, the second stage of plastic deformation, i.e., the recovery of the initial plastic deformation will be promoted.

For each quenching test, the residual stress along the thickness in the width direction was measured using the CCM, as shown in Figure 6. The results show that interrupted quenching can satisfactorily control the residual stress. The maximum tensile (core)/compressive (surface) residual stress was reduced to 184/199 MPa using interrupted quenching compared with the regular quenching, which produced a residual stress of 229/268 MPa. The compressive residual stress decreased by 42 MPa in the core and the tensile residual stress decrease by 72 MPa in the surface.

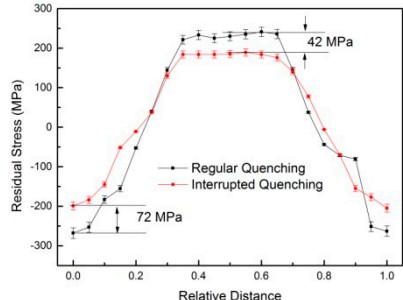

**Figure 6.** Comparisons of the measured residual stress in different quenching processes using the CCM.

In the interrupted quenching process, the cooling path changes only at temperatures less than 200 °C, which are outside the quench-sensitive temperature range. Therefore, the reduction in the cooling rate may not produce additional quench-induced coarse precipitation, which may lead to an additional performance loss after aging treatment. This lack of additional quench-induced coarse precipitation is illustrated in the comparison of the core and surface microstructures from both regular quenching and interrupted quenching processes, as shown in Figure 7. This Figure shows that no significant difference exists between the core and surface microstructures after regular quenching or interrupted quenching.

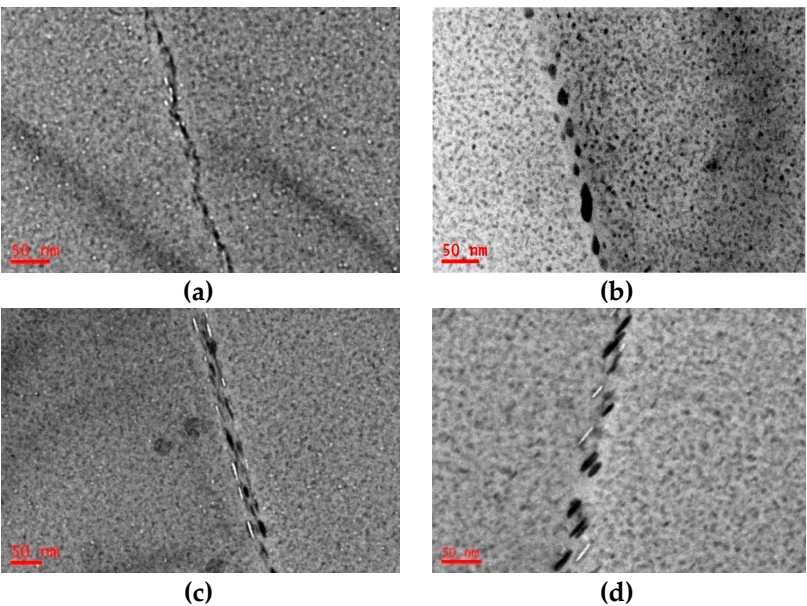

**Figure 7.** Comparison of the surface and core microstructures after either regular quenching or interrupted quenching: (**a**) surface and (**b**) core of regular quenching and (**c**) surface and (**d**) core of interrupted quenching.

In fact, the change in the cooling process at temperatures less than the quench-sensitive temperature range may not significantly affect the as-aged performance. The temperature changes during quenching are usually rapid, and the volume fraction of the accumulated precipitation (clustering) during quenching is limited compared to the amount of precipitation during the subsequent aging treatment. This precipitation is significantly enough to harden the as-quenched material, but does not affect the as-aged condition to any significant extent. As shown in Figure 8, the as-aged performance including the yield strength and electrical conductivity is almost the same after interrupted quenching and regular quenching. Therefore, in contrast to regular quenching, the interrupted quenching process reduced the residual stress without creating detrimental effects in the as-aged performance.

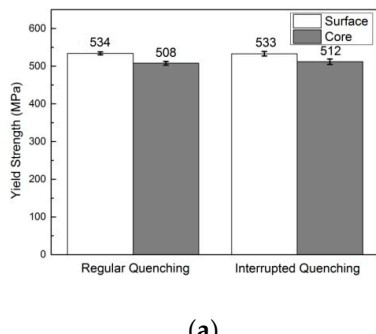
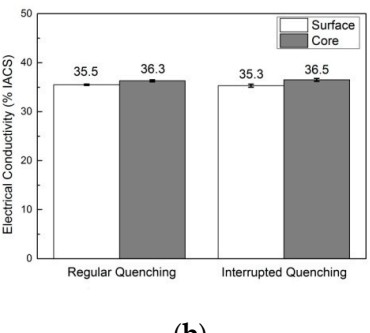

(**a**)                                                                      (**b**)

**Figure 8.** (**a**) Yield strength and (**b**) electrical conductivity of regular quenching and interrupted quenching after T74 aging treatment.

Furthermore, the Vickers hardness along thickness direction (in the middle-part of samples) after ageing treatment was shown in Figure 9. Results show that the interrupted quenching does not bring additional performance after ageing, which means that the hardenability will not deteriorate with interrupted quenching.

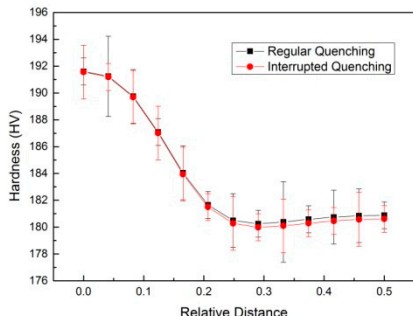

**Figure 9.** Comparisons of the hardness between regular quenching and interrupted quenching after T74 ageing treatment.

## 4. Conclusions

In this study, an interrupted quenching technology was proposed to reduce the residual stress using this phenomenon, following the idea that decreasing the yield strength in ambient temperatures during quenching can promote the recovery of initial plastic deformation and thus reduce the final residual stress.

The yield strength of the as-quenched 115 mm thick 7050 aluminum plate exhibited a significantly higher yield strength due to precipitation (solute clusters) during quenching. The quenched-in clusters can be partly eliminated after heating; thus, the yield strength will be decreased with heat preservation at higher temperatures. Interrupted quenching experiments were conducted in a 115 mm thick 7050 aluminum plate, wherein the spray quenching was stopped for 20 s after the initial 30 s of rapid cooling. The residual stress, yield strength, and electrical conductivity was measured compared with that of regular quenching.

Results show that the residual stress in the interrupted quenching process was 194/199 MPa. The residual stress was significantly reduced compared to the residual stress after regular quenching, which was 229/268 MPa. Moreover the yield strength and electrical conductivity for both surface and core was not decreased compared to those of regular quenching. It can be concluded that the interrupted quenching can reduce the residual stress significantly without additional (as-aged) performance loss.

**Author Contributions:** S.Y. was the principal writer, acquisition, analysis and interpretation of data for the work. The design of this work was provided by K.C., C.Z. and S.Y.

**Funding:** This research was funded by National Natural Science Foundation of China grant number [51327902], National Key Research and Development Program of China [No. 2016YFB0300801] and State Key Laboratory of High Performance Complex Manufacturing of Central South University (No. ZZYJKT2017-02).

**Conflicts of Interest:** The authors indicated no conflict of interest.

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
