# Peer review of "A New Path of Quench-Induced Residual Stress Control in Thick 7050 Aluminum Alloy Plates"

_metals, doi:10.3390/met9040393_

Reviewer 1 Report

How many compression tests were done? Please specify if the presented results are representative. For instance, bar graphs with YS values may be used with thestandard deviation indicated.

Author Response

1.      - How many compression tests were done? Please specify if the presented results are representative. For instance, bar graphs with YS values may be used with thestandard deviation indicated.

Response:  There are three tests was conducted per each value and take the average as the yield strength. And the bar graphs of the YS values includes thestandard deviation was provided, instead of old bar graphs.

Reviewer 2 Report

About the whole article
I understood that I am considering a new cooling process.
However, since the heat pattern is not shown, the reader can not imagine what kind of processing is being done.

Please indicate the intent of this cooling process from equilibrium state diagram and cooling curve of 7000 series.

Although the process of this time does not make use of the precipitation effect, please indicate the change of the hardness by it.

I think that the cooling rate will change with water flow in spray cooling, but how do you measure the temperature during cooling?

I think that X-ray is generally used for residual stress measurement, please tell me the reason why it was not used this time.

Is the part shown in the photo of Fig. 7 vacancy? Or is it a precipitate?

Author Response

1.      - About the whole article. I understood that I am considering a new cooling process. However, since the heat pattern is not shown, the reader can not imagine what kind of processing is being done. Please indicate the intent of this cooling process from equilibrium state diagram and cooling curve of 7000 series.

Response:  The hardenability (the precipitation difficulty) of alloy are usually be evaluated using the time-temperature-property (TTP) curve which expressed as the critical time required to precipitate a constant amount of solute. The TTP curve of 7050 aluminum alloy was shown in Fig.5., corresponding to the 99.5 % maximum hardness that approximately reflect the critical time of precipitation begin [5]. According to the TTP curve of AA7050, a high cooling rate must be adopted during the initial quenching to ensure the hardenability of the material, and the adjustment was made to the cooling rate only in the latter stage of quenching to avoid additional deterioration of final performance.

Fig. 5. The sketch of the regulation window of quenching process

2.      - Although the process of this time does not make use of the precipitation effect, please indicate the change of the hardness by it.

Response: The hardness was measured along the thickness direction per 5mm in center of the rectangle sample, which performed on a Vickers Hardness Tester HV-50 produced by Shanghai union Test Equipment Co., Ltd, and calibrated according to GB/T 231.2-2012. Hardness sample is stored at a temperature below 25℃.

3.      - I think that the cooling rate will change with water flow in spray cooling, but how do you measure the temperature during cooling?

Response: The temperature was not measured (except the surface temperature in interrupting period, because the core temperature measurement requires drill a hole to place the thermo-conductor, this destructive operation may affects the accuracy of CCM measurement of residual stress.

In this article, the cooling curves were provided finite element analysis (FEA) whose heat transfer analysis are consistent with the parameters in literatures [6], the accuracy of heat transfer parameters was certified in measurement. Mostly, the cooling curves in our study is used to express the idea of interrupted quenching, the residual stress or final performance are not derived from the heat transfer analysis. Means that the accuracy of these cooling curve is not as critical as your considered. We can not guarantee that the provided cooling curves is totally accurate, but the error is limited. But we can  guarantee that the general trend of temperature evolution during interrupted quenching is correct.

[6] Zhang J., Deng Y.L., Yang W. Design of the multi-stage quenching process for 7050 aluminum alloy. J. Mater. Design. 2014, 56 (4), 334–344.

4.      -I think that X-ray is generally used for residual stress measurement, please tell me the reason why it was not used this time.

Response: X-ray is only available for thin plates because the X-ray penetration depth is limited (<10 mm).

5.      - Is the part shown in the photo of Fig. 7 vacancy? Or is it a precipitate?

Response: It is the precipitates in grain boundary. For the studies in quench-in precipitates, the most different precipitates in different quenching condition is the precipitates in grain boundary.

Reviewer 3 Report

1- Line 202: 150 mm or 150 microns? Please check.

2- Figures 4 and 8 need uncertainty values.

3- What are those precipitates in figure 7 (black and white precipitates)?

4- The authors claim that there are some nano-size precipitates in the matrix that are dissolved when stopping or interrupting the quenching because the surface temperature rises to 150oC. This matter can be investigated further by means of  a differential scanning calorimetry (DSC). 

Author Response

Reviewer #3

1.- Line 202: 150 mm or 150 microns? Please check.

Response: 150 nm.

2- Figures 4 and 8 need uncertainty values.

Response: The uncertainty values are added.

3- What are those precipitates in figure 7 (black and white precipitates)?

Response: Both black and white precipitates are the inhomogeneous precipitates   phase (MgZn2) in grain boundary. White precipitates are results in the exfoliation of hardened phase during sample preparation. Further, for the studies in quench-in precipitates, the most different precipitates in different quenching condition is the precipitates in grain boundary.

4- The authors claim that there are some nano-size precipitates in the matrix that are dissolved when stopping or interrupting the quenching because the surface temperature rises to 150oC. This matter can be investigated further by means of  a differential scanning calorimetry (DSC).

Response: For the nano-size “precipitates”, we called as solute clusters. They actually are the local aggregation of solute atoms by vacancy transition (by solute-vacancy complex) rather than a stability structure as precipitation [2]. They were more like to short-range ordering rather than the long-range ordering of larger precipitates. Their quantitative can be analysis using in situ SAXS [1] or 3-dementional atom probe [3, 4] but our institute can not provided. We have not found any literature on cluster quantitative analysis by DSC, perhaps the accuracy of DSC analysis does not support such an unstable “precipitates”.

 [1] P. Schloth, A. Deschamps, C.A. Gandin, J.M. Drezet, Modeling of GP(I) zone formation during quench in an industrial AA7449 75mm thick plate. J. Mater. Design. 2016, 112, 46-57.

[2] Q. Zhao, Cluster strengthening in aluminium alloys. Scripta. Mater. 84-85(8) (2014) 43-46.

[3] Gang Sha, Alfred Cerezo, Early-stage precipitation in al–zn–mg–cu alloy (7050), Acta. Mater. 52(15) (2004) 4503-4516.

[4] S.P. Ringer, K. Hono, T. Sakurai, I.J. Polmear, Cluster hardening in an aged Al-Cu-Mg alloy, Script. Mater. 36 (5) (1997) 517-521.

Reviewer 4 Report

First I do not know if this manuscript corresponds to a re-submission, but it is very confusing for its review to show corrections as many sentences were deleted and other ones are new.

Instead of using “ambient temperature” use the term “room temperature”.

Some references in the text are noted using numbers, for example in line 58 “[6]”, but in other places are noted as names, for example in line 84 “Schloth”. Use a uniform nomenclature.

Line 35: With aging mechanical properties are not improved, the yield stress (or hardness) is reduced and the toughness is increased. Authors can point that “toughness is improved”.

Line 43: At the beginning of section 1 you talk about quenching and aging… but in line 43 you do not mention the aging process.

Line 48: “Finial”? means “Final”.

Line 54-58: You write several lines long sentence. Please make short sentences, as this will help readers to follow your ideas.

Lines 85-88: Italic characters are mixed with regular ones?

Line 137: Instead of using “the length of each cylindrical sample was parallel…” use the “longitudinal axis of each sample was parallel…”

Lines 155-160: This paragraph is not clearly written. If you wrote that you reach room temp. why then you said that the surface heats up again… then you do not reach room temperature on you sample.

Lines 140 and 192: You inform about the thermal treatments used in line 140 but in lines 192 you wrote that other 3 thermal treatments will be applied to samples. This must be clarified.

Lines 200-205. It is not clear how TEM samples where obtained and the zones of the original samples used for getting this TEM samples.

Lines 214-217. This paragraph/sentence should be rewritten because it is a confusing sentence and it is no clear how parameters used for computer simulations were obtained.

Line 225: in different state were “provided”? I think that is better “tested” or “studied”, as shown in figure 4(a). The first sentence must be split in two sentences.

Line 231: During the paused thermal treatment… you said that you stop the quench for about 20 seconds, and you leave in air… but now you write that you re-heat it up to 160º… this is confusing… reheated or leaved at room temperature?

Line 232: The resulting sentence of removing two paragraphs must be reviewed as it is too long and confusing.

Line 248-250: Sentence is not finished.

Line 255-57: Rewrite the sentence.

Line 258-260: You are writing about the thermal conductivity but figure 4b illustrates the evolution of electro-conductivity!!

Line 286 and 288… the temperature of the surface/reheating is 160, 140 or 150? In line 148 you give the furnace temperature with a precession of 2ºC (Probably too precise for a so big part). Now the temperature is more critical, and you cannot give a precise value.

Lines 300-304: You conclude that the reduction on residual stresses is related with the interrupted quenching technique… but what happens if you use a thinner of thicker part? May results change? What about error bars in residual stresses measurement? In this case differences are not so big, so it is important to look if there is any overlap between error bars.

Lines 344. The values of 179/172 correspond to yield strength and to no residual stresses. You are comparing yield stresses with residual stresses and this has no sense.

The most important part of the manuscript are the conclusion and are not well supported by the experimental results.

Author Response

1.      - Instead of using “ambient temperature” use the term “room temperature”.

Response: For example “In this paper, a new path is proposed that reduces the residual stress through decreases the yield strength in ambient temperatures by eliminating the precipitation hardening effect during quenching”. The residual stress actually formed at the intermediate temperatures during quench, not at the room temperature, the term “ambient temperature” means the variant temperatures during quenching. “The yield strength in ambient temperatures” means the yield strength with temperature dependency. "Room temperature" refers to the temperature after quenching, which is the same as the ambient temperature of air (about 25°C).

2.      - Some references in the text are noted using numbers, for example in line 58 “[6]”, but in other places are noted as names, for example in line 84 “Schloth”. Use a uniform nomenclature.

Response: The uniform nomenclature was modified.

3.      - Line 35: With aging mechanical properties are not improved, the yield stress (or hardness) is reduced and the toughness is increased. Authors can point that “toughness is improved”.

Response: In the case of the same quenching condition, the yield stress (or hardness) is reduced and the toughness is increased taken the comparison of different ageing condition into account. But this study is focus on quenching condition, the deteriorate of mechanical properties by the coarse inhomogenous precipitation not only may decrease the yield stress (or hardness) but also toughness. So your suggestion is not applied.

4.      - Line 43: At the beginning of section 1 you talk about quenching and aging… but in line 43 you do not mention the aging process.

Response: Although it is well known that quenching is usually followed by ageing treatment, but the topic of this article is only discussed on quenching process. Thus, aging process was not mentioned in latter section.

5.      - Line 48: “Finial”? means “Final”.

Response: Misspelling, corrected.

6.      - Line 54-58: You write several lines long sentence. Please make short sentences, as this will help readers to follow your ideas.

Response: Origin text: “Furthermore, the material will harden by the precipitation phenomena that occur during quenching, which produce a higher yield stress at ambient temperatures during quenching that will allow larger residual stress magnitudes to be supported [7]. This hardening effect is especially significant in the quenching of thick plates, such as the surface yield strength results regarding as-quenched thick AA7449 plates, wherein the 75 mm thick plate exhibited a much larger yield strength (255 MPa) than the 20 mm thick plate (210 MPa) [8]. These additional hardening effects make it more difficult to reduce the residual stress by lowering the cooling rate.

Changed text: “Furthermore, the material will harden by the precipitation phenomena that occur during quenching. Hardening during the quench will give rise to a higher yield strength at ambient temperatures during quenching, which in turn will produce larger residual stress magnitudes [7]. This hardening effect is especially significant in the quenching of thick plates. For example, the yield strength of as-quenched thick AA7449 plates, wherein the 75 mm thick plate exhibited a much larger yield strength (255 MPa) than the 20 mm thick plate (210 MPa) [8]. These additional hardening effects make it more difficult to reduce the residual stress by lowering the cooling rate.”

7.      - Lines 85-88: Italic characters are mixed with regular ones?

Response: Format error, corrected.

8.      - Line 137: Instead of using “the length of each cylindrical sample was parallel…” use the “longitudinal axis of each sample was parallel…”

Response: Origin text: “the length of each cylindrical sample was parallel to the width direction” change to “longitudinal axis of each sample was parallel to the width direction”.

9.      - Lines 155-160: This paragraph is not clearly written. If you wrote that you reach room temp. why then you said that the surface heats up again… then you do not reach room temperature on you sample.

Response: In interrupting, the heat transfer from surface to circumstance decrease significantly, and the surface material will be re-heated by the heat transfer from the internal to surface.

10.  - Lines 140 and 192: You inform about the thermal treatments used in line 140 but in lines 192 you wrote that other 3 thermal treatments will be applied to samples. This must be clarified.

Response: The quenching tests (using rectangle rectangular samples )was includes regular quenching and interrupted quenching. The compression tests (using small samples cutted from as-quenched rectangle rectangular samples) was achieved after respectively for regular quenching and interrupted quenching, includes 3 heat treatment status includes as-quenched, re-solutioned, reheated to 160 °C for 20 s.

11.  - Lines 200-205. It is not clear how TEM samples where obtained and the zones of the original samples used for getting this TEM samples

Response: Slices cut from cylinder sample of surface and core (cut section is perpendicular to the longitudinal axis of each sample), respectively. Samples were mechanically ground to a thickness of approximately 150 nm from which discs of 3 mm diameter were punched.

12.  - Lines 214-217. This paragraph/sentence should be rewritten because it is a confusing sentence and it is no clear how parameters used for computer simulations were obtained.

Response: These paragraph was rewritten.

13.  - Line 225: in different state were “provided”? I think that is better “tested” or “studied”, as shown in figure 4(a). The first sentence must be split in two sentences.

Response: Modified as “In this study, samples from the surface and core in different states were studied, as shown in Fig. 4.(a). The heat treatment status for each sample is shown in Table 2”.

14.  - Line 231: During the paused thermal treatment… you said that you stop the quench for about 20 seconds, and you leave in air… but now you write that you re-heat it up to 160º… this is confusing… reheated or leaved at room temperature?

Response: I recognized that the corresponding description may confusing readers, the description has been modified to: “That is, after quenching for a period of time (30 s), the surface was cooled to approximately 100 °C and the core was cooled to about 200 °C (below the quench-sensitive temperature range). Then, the spray quenching was interrupted for a period of time (about 20s). In interrupting, the heat transfer from surface to circumstance decrease significantly, and the surface material will be re-heated by the heat transfer from the internal to surface. Finally, the material was cooled to room temperature through spraying.”

15.  - Line 232: The resulting sentence of removing two paragraphs must be reviewed as it is too long and confusing.

Response: These paragraph was rewritten.

16.  - Line 248-250: Sentence is not finished.

Response: This sentence is deleted.

17.  - Line 255-57: Rewrite the sentence.

Response: These paragraph was rewritten.

18.  - Line 258-260: You are writing about the thermal conductivity but figure 4b illustrates the evolution of electro-conductivity!!

Response: Sorry for such an obvious mistake, and corrected.

19.  - Line 286 and 288… the temperature of the surface/reheating is 160, 140 or 150? In line 148 you give the furnace temperature with a precession of 2ºC (Probably too precise for a so big part). Now the temperature is more critical, and you cannot give a precise value.

Response: The temperature of the surface/reheating is 141 °C which measured using a contact thermocouple. Furnace temperature is corresponding to the temperature of solution treatment (before quenching). The temperature was not measured (except the surface temperature in interrupting period, because the core temperature measurement requires drill a hole to place the thermo-conductor, this destructive operation may affects the accuracy of CCM measurement of residual stress.

In this article, the cooling curves were provided finite element analysis (FEA) whose heat transfer analysis are consistent with the parameters in literatures [6], the accuracy of heat transfer parameters was certified in measurement. Mostly, the cooling curves in our study is used to express the idea of interrupted quenching, the residual stress or final performance are not derived from the heat transfer analysis. Means that the accuracy of these cooling curve is not as critical as your considered. We can not guarantee that the provided cooling curves is totally accurate, but the error is limited. But we can  guarantee that the general trend of temperature evolution during interrupted quenching is correct.

[6] Zhang J., Deng Y.L., Yang W. Design of the multi-stage quenching process for 7050 aluminum alloy. J. Mater. Design. 2014, 56 (4), 334–344.

20.  - Lines 300-304: You conclude that the reduction on residual stresses is related with the interrupted quenching technique… but what happens if you use a thinner of thicker part? May results change? What about error bars in residual stresses measurement? In this case differences are not so big, so it is important to look if there is any overlap between error bars.

Response: The residual stress can be reduced in thin plates (T<50 mm), but the mechanism of residual stress reduction is the decrease of thermal stress, not the decrease of hardened effect that inferred in our study. As mentioned in backgrounds, “Compared with regular quenching, multistage quenching can reduce the maximum residual stress by only approximately 5-10%”. As shown in below Figure, the residual stress is sensitive to spraying condition (affects the heat-transfer coefficient) in thickness less than 50 mm, but not for thicker plates. For thin plates, the residual stress can be reduced through several technology such as using the warm water as the quenchant. So far, the residual stress can not reduced significantly for thick plates because the spraying condition is not as sensitive as for thin plates. And these is the innovation value of interrupted quenching technology.

Further, the error bar was added.

21.  - Lines 344. The values of 179/172 correspond to yield strength and to no residual stresses. You are comparing yield stresses with residual stresses and this has no sense.

Response: It is a mistake, and corrected to 199/184 MPa(corresponding to the residual stress of intterupted quenching)

22.  - The most important part of the manuscript are the conclusion and are not well supported by the experimental results.

Response: The conclusion was changed to:

“In this study, an interrupted quenching technology was proposed to reduce the residual stress using this phenomena, following the idea that decreasing the yield strength in ambient temperatures during quenching can promote the recovery of initial plastic deformation and thus reduce the final residual stress.

The yield strength of the as-quenched 115 mm thick 7050 aluminum plate exhibited a significantly higher yield strength due to precipitation (solute clusters) during quenching. The quenched-in clusters can be partly eliminated after heating; thus, the yield strength will be decreased with heat preservation at higher temperatures.. Interrupted quenching experiments were conducted in a 115 mm thick 7050 aluminum plate, wherein the spray quenching was stopped for 20 s after the initial 30 s of rapid cooling. The residual stress, yield strength, and electro-conductivity was measured compared with that of regular quenching:

 Results show that the residual stress in the interrupted quenching process was 199/184 MPa. The residual stress was significantly reduced compared to the residual stress after regular quenching, which was 244/245 MPa. And the yield strength and electro-conductivity for both surface and core was not decreases compare to these of regular quenching. Concludes that the interrupted quenching can reduce the residual stress significantly without additional (as-aged) performance loss.”

Round  2

Reviewer 2 Report

I think that it is good because the pointed out item is corrected.

Author Response

Mar 19, 2019 Dear reviewer:       We sincerely thank the editors and all reviewers for their valuable feedback that we have used to improve the quality of our manuscript “A New Path of Quench-Induced Residual Stress Control in 7050 Aluminum Alloy Thick Plates” (Manuscript ID: Metals-458880(441148)). Those comments are helpful for us to revise and improve our paper. Your best wishes! Sincerely, The co-authors of this paper

Reviewer 4 Report

Authors answer referees comments properly. A few improvements could be made in the text.

Line 248-249: The last sentence can be rewriteen as: “The compressive residual stress decreased by 42 MPa in the core and the tensile residual stress decrease by 72 MPa in the surface.” Adding the terms “compressive” and “tensile” make the sentence much clearer.

Lines 277-280: You refer to figure 5(b) and to the residual stresses again in figure 5(b). This is wrong as figure 5 shows the thermal evolution. First time that you mention figure 5(b) in this paragraph you are referring to figure 9 and the seconta time perhaps to  Figure 8(b)? Please doublecheck this!!

Author Response

Mar 19, 2019Dear reviewer:

       We sincerely thank the editors and all reviewers for their valuable feedback         that we have used to improve the quality of our manuscript “A New Path of         Quench-Induced Residual Stress Control in 7050 Aluminum Alloy Thick Plates”        (Manuscript ID: Metals-458880(441148)). Those comments are helpful for us to        revise and improve our paper.

1.      - Line 248-249: The last sentence can be rewriteen as: “The compressive residual stress decreased by 42 MPa in the core and the tensile residual stress decrease by 72 MPa in the surface.” Adding the terms “compressive” and “tensile” make the sentence much clearer.

Response: Thanks for your suggestion,  and the make amendments accordingly.

2.      - Lines 277-280: You refer to figure 5(b) and to the residual stresses again in figure 5(b). This is wrong as figure 5 shows the thermal evolution. First time that you mention figure 5(b) in this paragraph you are referring to figure 9 and the seconta time perhaps to  Figure 8(b)? Please doublecheck this!!

Response: Line 277-280 was doublechecked. Thank you for your caraful examination. And the sentence was corrected to : “Furthermore, the Vickers hardness along thickness direction (in the middle-part of samples) after ageing treatment was shown in Fig. 9”

This manuscript is a resubmission of an earlier submission. The following is a list of the peer review reports and author responses from that submission.

Round  1

Reviewer 1 Report

The paper concerns a very interesting topic. However several remarks should be taken into consideration.

First of all. Please check the English. The help of native English is recommended.

The detailed description of the test stand for quenching by spraying is recommended. Otherwise, the reproduction of your results by other scientist renders difficulties.

Which part of the aluminum plate was used for cutting out the specimens for compression tests? It is stated that they are cut out from the middle of the thickness, but was it left, the right of the center part of the plate?

How many compression tests were done? Please specify if the presented results are representative. For instance, bar graphs with YS values may be used with thestandard deviation indicated.

Were the temperatures during quenching measured for instance by K-type thermocouple or only FEM modeled?

Is the fig.3 consistent with real cooling curves? Since the heat transfer coefficient depends on surface roughness, spraying nozzle type, nozzle to surface distance and many other unpredictable factors is it proper to adopt the values needed for FEM modeling on the basis of the literature studies?

Was the flow rate measured by flowmeter? Was the flow equally distributed over the cooled area?

Please specify exact places of hardness measurement Was it the middle part of the plate widht?

What was the cooling agent during the experiment? In line 177 it is stated to be water. Was the Lidenfrost effect taken into consideration during FEM modeling?

Check the figures referenced in the text. For instance, in L 200 it is stated the reader should check figure 4a. There is no fig. 4a

The literature studies may be updated. Half of the position used is older than 10 years.

Reviewer 2 Report

This article really needs a thorough proof read from an English speaker - there are a lot of grammatical errors which makes the document difficult to read.

Suggesting a change to the heat treatment of an alloy such as this would require extensive testing to validate the effect on final properties. Ideally, mechanical tests should be undertaken including tensile testing, fracture toughness and SCC testing. There was no data given related to electrical conductivity which is usually used in these alloys to infer corrosion resistance. These results could be backed up with microscopy analysis including optical and, ideally TEM.